# History of the Biodiversity of Ladybirds (Coccinellidae) at the Black Sea Coast of the Russian Caucasus in the Last 120 Years—Does the Landscape Transformation and Establishment of *Harmonia axyridis* Have an Impact?

**DOI:** 10.3390/insects11110824

**Published:** 2020-11-23

**Authors:** Andrzej O. Bieńkowski, Marina J. Orlova-Bienkowskaja

**Affiliations:** A.N. Severtsov Institute of Ecology and Evolution, Russian Academy of Sciences, 119071 Moscow, Russia; bienkowski@yandex.ru

**Keywords:** Coccinellidae, ladybugs, Caucasus, Coleoptera, beetles, harlequin ladybird, subtropics, biological invasions, alien species, exotic species

## Abstract

**Simple Summary:**

Studies of the history of regional insect fauna are important for understanding the changes in ecosystems and are therefore crucial for conservation decisions. The harlequin ladybird is a global invader that causes the decline of native ladybirds in some countries. Therefore, it is advisable to monitor the ladybird fauna in regions recently occupied by this species. We analyzed the dynamics of the fauna at the main sea resort of Russia over a period of 120 years to determine the following: (1) how the ladybird biodiversity changed during the intensive landscape transformation; (2) what alien species introduced for pest control have occurred to date; and (3) what the impact is of the harlequin ladybird on the ladybird fauna. We examined specimens collected by us and 54 other collectors including specimens from old museum collections and reconstructed the history of the biodiversity like a picture from puzzle pieces. Surprisingly, landscape transformation did not cause a decrease but rather an increase in ladybird biodiversity; most of the species recorded before 1930 have occurred to date, and 23 other species have spread to the region. Four released alien species occur in gardens and streets but not in natural habitats. The harlequin ladybird has been abundant for 10 years, but no ladybird species have disappeared in the region because of it.

**Abstract:**

Studies of the history of regional insect fauna are important for understanding the changes in ecosystems. We analyzed the dynamics of ladybird fauna at the main sea resort of Russia over a period of 120 years to determine the following: (1) what species disappeared and what species appeared during landscape transformation; (2) what alien species introduced for pest control have been observed to date; and (3) whether the establishment of the global invader *Harmonia axyridis* (Coccinellidae) caused the extinction of some ladybird species in the Caucasus. We examined specimens collected by us and 54 other collectors including specimens from old museum collections and detected 62 species, 50 of which were collected in recent years (2011–2020). Landscape transformation and recreational use have caused not a decrease but an increase in ladybird biodiversity. Twenty-nine of 34 species recorded before 1930 have been observed in the region to date. Twenty-three other species have spontaneously spread to the region between 1930 and 2020 because of the creation of suitable anthropogenic habitats or because of unintentional introduction. *Rodolia cardinalis*, *Cryptolaemus montrouzieri*, *Lindorus lophanthae,* and *Serangium montazerii* were released for pest control, and they occur in gardens and streets but not in natural habitats. *Harmonia axyridis*, which appeared approximately 10 years ago, is abundant in urban and natural habitats, but there is no evidence that it caused the elimination of any ladybird species.

## 1. Introduction

Studies of the dynamics of regional flora and fauna (i.e., history of biodiversity over decades or centuries) are crucial for making many ecological and conservational conclusions [1]. The analysis of changes in the biodiversity of a particular taxonomic or ecological group in a particular region could reveal the impact of climate change and anthropogenic landscape transformation and the establishment of alien species [2]. The historical approach, i.e., the study of current dynamics of flora and fauna, is now becoming more and more popular in zoology and botany [3]. The accumulated information on localities of species collections (in particular, in the Global Biodiversity Information Facility [4]) clearly indicates that the ranges of plant and animal species are changing quickly; significant changes often occur in just several decades. Since the species live in the territories which have been intensively used and changed by humans over many centuries, we believe that the current regional flora and fauna are the product of the long-term interaction between nature and human society. Therefore, the history of biodiversity should be analyzed in the context of human history. 

The monitoring of regional insect fauna over centuries is rare. However, owing to the interest of professional and amateur entomologists, ladybirds (Coccinellidae) are among the few insect groups for which it is possible to collect reliable information on the biodiversity history [5]. Most Coccinellidae species were collected in the same microhabitats, namely, on leaves of trees, bushes, and grasses, and the main methods of collection were the same in the 19th, 20th, and 21st centuries, making the faunistic data obtained in different historical periods comparable.

The purpose of our work was to analyze the dynamics of the ladybird fauna at the Black Sea coast of the Russian Caucasus during the last 120 years. The history of Coccinellidae fauna in this region is interesting for three reasons. First, drastic changes have occurred in this region since 1900 because of the development of the main sea resort of Russia [6]. Therefore, it is of interest to determine what ladybird species disappeared and what species appeared in the face of strong landscape changes in different periods of the resort development. Second, several species of alien Coccinellidae were intentionally introduced to the region for pest control in different decades of the Soviet period [7]. Therefore, it is of interest to assess which of these species still occur in the region. Third, the fauna of Coccinellidae could potentially change because of the impact of the recently-established alien ladybird *Harmonia axyridis* (Pallas, 1773) [8,9,10].

*Harmonia axyridis* is native to East Asia [11] and has been released worldwide for biological control of aphids for approximately 100 years [12], becoming a global invader [12]. This species poses a threat to the biodiversity of aphidophagous insects, particularly coccinellids, through competition and predation [12]. It has caused the decline of native ladybirds in many regions of the world [13,14]. Currently, it is important to monitor the Coccinellidae fauna in the regions recently occupied by this invader [15]. The Black Sea coast of the Russian Caucasus is perfect for such monitoring because the invasion of *H. axyridis* into the region is well documented. In the early 2000s, established populations of *H. axyridis* were found in Western Europe, and the quick spread of this species across Europe began [16]. In 2011, the dispersal wave of this alien ladybird from Western Europe to the east reached the Caucasus, and *H. axyridis* became the most abundant species of Coccinellidae in the region [8,9,10,17].

The study of Coccinellidae fauna on the Black Sea coast of the Russian Caucasus began in the late 19th century. The first inventories of Coccinellidae of the region were published in 1915 [18] and 1916 [19]. Then, no studies on Coccinellidae of the region were published for almost 100 years. The available literature on this subject is scarce; the list of 11 ladybird species collected in the vicinity of Sochi in 2011 and 2012 was compiled [20], and several articles were devoted to ladybird species released for pest control and the invasion of *H. axyridis* to the region [8,20,21,22,23,24,25].

Fortunately, Coccinellidae is a very popular group attracting the attention of people, and the collection of ladybirds does not require special methods. Many entomologists have visited the resorts of the Black Sea coast of the Russian Caucasus and collected ladybirds there. These specimens have been accumulated in collections. In particular, we found approximately two hundred specimens collected in the late 19th and early 20th centuries in the collections of the Zoological Institute of Russian Academy of Sciences (St. Petersburg) and the Zoological Museum of the Moscow State University (Moscow). Thousands of specimens were collected in the subsequent periods and deposited into different museums and private collections. These accumulated data have never been integrated or analyzed. Neither the current ladybird species composition nor the dynamics of fauna have been studied. The ecological consequences of the establishment of *H. axyridis* in the region have not yet been studied.

The aim of our study was to integrate all information about the history of Coccinellidae fauna in the last 120 years from museums and private collections including our own collection and to answer the following questions:What ladybird species occurred in the region in late 19th and early 20th century and then disappeared during the development of the resort?What species appeared in the region in the period of landscape transformation connected with resort development?What ladybird species intentionally introduced for pest control in the Soviet period still occur in the region?Did establishment of *H. axyridis* cause extinction of some ladybird species in the region?

## 2. Materials and Methods

### 2.1. The Study Region

The study region is the Black Sea coast of the Northern Caucasus from 45.06 N, 37.13 E in the northwest to 43.40 N, 40.01 E in the southeast (Figure 1). This region is situated in Krasnodar Territory in the south of European Russia and includes the Anapa, Novorossijsk, Gelendzhik cities, Tuapsinskiy district and Sochi city, the main sea resort region of Russia. Up to 10 million people from all over the country spend their holidays there every year [6].

This region is the band of coastal and foothill areas between the sea and the crest of the Great Caucasian ridge. The length of this band is approximately 270 km, the width ranges up to 30 km, and the area is 8913 km^2^. Elevations in the localities of the ladybird collection were up to 1100 m above sea level. The climate ranges from dry Mediterranean in the northwest of the region to wet subtropical in the southeast [26]. The Black Sea coast of the Caucasus is protected from northern winds by mountains, so the climate here is much warmer than in adjacent regions. The region is a landscape island that differs strongly from the surrounding regions and has a high level of plant and animal endemism [27].

In the early 20th century, the landscape in the region was rural, with native foothill vegetation, villages, gardens, and private estates with parks [6]. Many swamps were found in river valleys and coastal regions. In the 1920s, the quick development of the all-Soviet resort began, and by the late 1930s, the landscape had changed drastically. The swamps were irrigated, sanatoriums were built in the place of private estates, and the villages turned into resort towns. The railway and motorways were constructed, and traffic flow became intense. During the Soviet period, the vegetation changed drastically because, for several decades, plants from all over the world were planted in the streets and parks of the resort towns. Many of these plant species spread to natural habitats and became invasive [28]. The last period of landscape transformation in the region was connected with the construction of Olympic facilities and roads during the preparation for the Sochi Olympic Games in 2014. Many alien insect species, including invasive pests, were unintentionally introduced to the region in this period with the imported seedlings [29].

### 2.2. Methods of Material Collection and Integration of Information

We collected 1710 specimens of ladybirds in different localities of the region in 1987, 1990, 1992, 1995, 2008, 2013, 2014, 2016, 2017, 2018, and 2020. All these specimens are in the collection of the first author. The main method of collection was sweep netting of grasses and shaking of branches of trees and shrubs during both day and night. An exhauster with a rubber bulb and replaceable tanks was used for the collection of small specimens from the net [30].

We also examined specimens collected by other entomologists and deposited in the Zoological Institute of the Russian Academy of Sciences, the Zoological Museum of the Moscow State University, the All-Russian Center for Plant Quarantine, and the Scientific Center of Zoology and Hydroecology of the National Academy of Sciences of Armenia as well as specimens collected by 18 private collectors (see Acknowledgments). Overall, specimens collected by at least 56 collectors for over 120 years were studied. The specimens were identified with the use of the identification guides [31,32,33].

In addition, we summarized all available published information about records of ladybirds in this region and examined six photographs from the iNaturalist website [34]. All the information associated with Coccinellidae records with geographical coordinates of each locality is integrated in the electronic Appendix A.

Our methods did not allow us to estimate the abundance of species. We detected only the presence of the species in the region in the particular year. The intensity of the study of ladybird fauna in the region in different decades is shown in Figure 2.

## 3. Results

### 3.1. Species Recorded in the Early Period of Resort Development

Thirty-four ladybird species were recorded on the Black Sea coast of the Russian Caucasus in 1888–1927, i.e., in the early period of resort development (Table 1). Twenty-nine of these species (85%) were also recorded in the region between 1990 and 2020. None of the species that were common in 1888-1927 disappeared later. *Coccinula sinuatomarginata* was found in 1910, 1911, and 1912 in three districts in the northeast of our study region but was never found later. This species is rare in European Russia [33] and other European countries [32]. The following four species were recorded once in 1888–1927 but were never recorded later—*Ceratomegilla notata* (recorded in 1910), *Coccinella hieroglyphica* (in 1911)*, Scymnus inderihensis* (in 1927), and *Sospita vigintiguttata* (in 1907). Only having one record of each of these species indicates that they were probably rare in the region.

### 3.2. The Species First Recorded in the Subsequent Periods of Resort Development

In 1940–2020, 23 other ladybird species were recorded in the region (Table 2; intentionally-released alien species are not counted here). 

In the study area, the average rate of recording new ladybird species is three per decade. From two to four new species were detected in each decade between the 1940s and the 2010s. The only exception was the 1990s. As many as eight new species were collected in the region in that decade. The collection intensity was low in the 1960s–1980s (Figure 2), so it is likely that the new species that appeared in the region over these 30 years could first be detected only in the 1990s. The possibility of a comparatively-high number of newly-recorded ladybird species in 1990s being just accidental cannot be excluded.

Once recorded in the region, many of the species in this group were then repeatedly collected in subsequent years. For example, *Adalia decempunctata* was not collected in 1888–1956 but was collected in the 11 years during the period 1957–2020 (Appendix A). This aphidophagous species is typical for European deciduous forests and could not have been just overlooked by the collectors before 1957 because it is not small (up to 5 mm) and because is brightly colored and easily seen on trees and bushes [35].

Another example is *Parexochomus nigromaculatus.* This xerophilous, mainly aphidophagous, species is widely distributed in the steppes and deserts of Europe, Middle Asia, and Northern Africa [35]. This species was not collected at the Black Sea coast of the Russian Caucasus in 1888–1969 but was then collected for nine years between 1970 and 2020 (Appendix A). It is extremely unlikely that the species was just overlooked for more than 70 years and then began to be noticed.

All species first recorded in the region in the 1940s–1980s except the rare species *Oenopia impustulata* were then repeatedly found in subsequent decades, indicating that the species became established in the region and that the populations existed for several decades.

### 3.3. Species Intentionally Introduced to the Region for Pest Control

The following alien Coccinellidae have been released in the Caucasus in the Soviet period for biological control of pests—*Harmonia axyridis* (Pallas, 1773) was released after 1927 in Georgia, then it was released at the Black Sea coast of Russia; *H. conformis* (Boisduval, 1835) was released after 1958 in Georgia; *H. dimidiate* (Fabricius, 1781) was introduced to different regions of the Caucasus in 1986 and 1990 and used in greenhouses; *Cryptolaemus montrouzieri* (Mulsant, 1853) was released after 1933 in Georgia, then it was widely released along Black Sea coast of the Caucasus; *Serangium montazerii* (Fürsch, 1995) was released in Georgia after 1973 under the name *S.*
*parcesetosum* (Sicard, 1929), then it was widely released along Black sea coast of the Caucasus; *Lindorus lophanthae* (Blaisdell, 1892) was released after 1947; *Rodolia cardinalis* (Mulsant, 1850) was released after 1931; *R. rufopilosa* Mulsant, 1850 was released after 1955 in Georgia only; *Chilocorus infernalis* (Mulsant, 1853) was released after 1973 only in Georgia; *C. inornatus* (Weise, 1887) was released in 1935 in Georgia (Abhazia) only; *C. rubidus* (Hope, 1831) was released after 1935; *C. geminus* (Zaslavskij, 1962) was released after 1963 in Georgia only; *Nephus reunioni* (Fürsch, 1974) was released before 1987, only in Georgia; and *Aiolocaria hexaspilota* (Hope, 1831) was released after 1966 only in Ukraine, Georgia, and Kazakhstan [7,36,37]. Our data have shown that the populations of five of these species still exist in the region (Table 3). Besides that, specimens of *C. kuwanae* (Silvestri, 1909) introduced from Japan were released [7], but we have revealed that *C. kuwanae* (Silvestri, 1909) is the junior synonym of *C. renipustulatus* (Scriba, 1790), which is the native element of the ladybird fauna at the Black Sea coast of Russia [38].

*Rodolia cardinalis* is native to Australia. This specialized predator of *Icerya purchasi* (Maskell, 1878) was widely used for control of this pest on citrus plantations in the Caucasus. The first specimens were delivered to the region in 1931, and the species quickly became established [39]. To date, almost all citrus gardens in the Black Sea coast of the Russian Caucasus have been cut down, and the fate of the *R. cardinalis* population has been unknown [22]. We tried to find this ladybird for several seasons by shaking branches of *Acacia dealbata* Link. infested with *I. purchasi*, but we were unsuccessful [22]. Finally, we found a local population of *R. cardinalis* in one of the last tangerine groves in the region, in the south of the Adler district of Sochi (43.409 N, 39.984 E). Two specimens of *R. cardinalis* were found in this garden in 2018, and one specimen was found in 2020. Obviously, *R. cardinalis* has become rare in the region because citrus trees have become rare.

*Cryptolaemus montrouzieri,* native to East Australia and New Caledonia, has been released in different regions of the Caucasus, including Sochi, many times since 1933 [37,39]. This species was regularly used for biological control of Pseudococcidae and Coccidae by the method of seasonal colonization on plantations of tea, grapes, citrus, and ornamental plants [21]. However, the specimens that were released died out in winter and early spring, and no established populations were recorded for several decades [7,36,39]. In 2010, breeding of *C. montrouzieri* was detected in Sochi, and in 2011 and 2012, larvae and pupae were abundant on *Nerium oleander* L. infested by Coccidae in Sochi [21]. According to our data, *C. montrouzieri* is now common in Sochi. In 2013, 2016, 2018, and 2020, 71 specimens of this species were collected on *Nerium oleander, Thuja* sp., *Hibiscus* sp., and other ornamental bushes and tangerine trees in six localities in the Central and Adler districts of Sochi (Appendix A).

*Lindorus lophanthae* (=*Rhyzobius lophanthae* Blaisdell, 1892), native to Australia, was released for the control of Diaspididae in 1948 in Georgia and became established and then spread in the Caucasus. The whole population of *L. lophanthae* in the Caucasus derives from laboratory culture, established from just one pair of insects collected in Italy near Rome [40]. There was no information about the state of the *L. lophanthae* population after the 1950s. We collected nine specimens of this species in 2016, 2017, 2018, and 2020, which means that the population still exists in the region. The specimens were collected by shaking bushes in the streets in the Central district of Sochi and in two localities of the Adler district in the evening and at night. One specimen was caught on the beach in flight.

*Serangium montazerii* is native to India, Pakistan, and Syria and was released in 1973 in the Caucasus (Abkhazia, other regions of Georgia, Azerbaijan, and Sochi) to control *Dialeurodes citri* (Ashmead, 1885) (Aleyrodidae) on citrus trees. Initially, *S. montazerii* became established and began to spread spontaneously [7], but most citrus gardens in the region have been cut down in recent decades, and there has been no information about the state of the *S. montazerii* population after the 1970s. We collected 11 specimens of this species in three localities in 2017, 2018, and 2020 and therefore confirmed that the population still exists. We found the specimens on bushes in the street and in one of the last tangerine groves in the region.

*Harmonia axyridis* is native to Siberia, Far East, the northeast of Kazakhstan, Mongolia, China, North Korea, South Korea, Japan, and the north of Vietnam [11] and has been released in the Caucasus many times since 1927 [36]. The first established population in the Caucasus was detected in 2011 in Sochi [23,41]. According to our data, *Harmonia axyridis* has become the most abundant and widespread ladybird in the region. This species was collected in all years of collection between 2012 and 2020 in all districts of the region of study. We collected the species not only in urban landscapes (on *Rosa, Ficus, Ligustrum, Nerium, Hibiscus, Spiraea, Lysimachia, Rubus,* and other ornamental bushes), but also outside the towns. In particular, it was abundant in the deciduous forests in the foothills in Sochi National Park on *Alnus* and other trees and along the rivers on *Salix*. More than 400 adults, larvae, and pupae of *H. axyridis* were collected.

### 3.4. Biodiversity of Coccinellidae before and after the Establishment of H. axyridis

*Harmonia axyridis* has been abundant in the region since 2011. To assess the possible impact of its establishment, we compared the lists of ladybird species detected in two periods of collection—1985–2004 (i.e., well before the first record of *H. axyridis* in the region) and 2016–2020 (i.e., in the period when *H. axyridis* had been abundant in the region for at least five years (Table 4)).

Thirty nine ladybird species were collected in 1985–2004, and thirty nine ladybird species were collected in 2016–2020. The intentionally-introduced species are not counted here.

Nine species were detected in 1984–2004 but not in 2016–2020. Seven of these species (*Coccidula scutellata*, *Parexochomus melanocephalus*, *Hyperaspis concolor*, *Oenopia impustulata*, *Myzia oblongoguttata*, *Calvia quindecimguttata*, and *Ceratomegilla undecimnotata*) are rare—they were recorded only 1–3 times in all 120 years of collection. *Bulaea lichatschovii* is not so rare. But interspecific competition between *Bulaea lichatschovii* and *Harmonia axyridis* is unlikely since *B. lichatschovii* is a phytophagous ladybird feeding mainly on Chenopodiaceae [35], while *H. axyridis* is aphidophagous [35]. Therefore, the absence of *B. lichatschovii* in the samples in 2016–2020 is not connected with the establishment of *H. axyridis*.

*Hippodamia tredecimpunctata* also was detected in 1985–2004 and was not detected in 2016–2020. This species occurs in swamps and wet meadows and near waterbodies [42]. It feeds mainly on aphids and powdery mildew on grasses [42]. Since *H. axyridis* also occurs in the same habitats and feeds on aphids, interspecific competition between these species is theoretically possible. However, it is difficult to confirm or refute this hypothesis because *Hippodamia tredecimpunctata* is rather rare. It was collected in 1910, 1952, 1982, 1986, 1987, and 1990 in different districts of the region that was studied (Appendix A). Further monitoring of the population of *Hippodamia tredecimpunctata* is advisable.

Therefore, to date, there has been no evidence of extinction of any ladybird species in the region because of the establishment of *H. axyridis*.

## 4. Discussion

### 4.1. General Biodiversity of Coccinellidae in the Region

The examination of ladybird specimens collected by dozens of amateur and professional entomologists and the integration of all these data have provided a good picture of Coccinellidae fauna of the region and the dynamics of this biodiversity from the early 20th century until now. In total 62 ladybird species were recorded, representing 61% of all ladybird species recorded in southern European Russia [33]. Comparison with other regions of the Caucasus shows that the fauna of ladybirds of the Black Sea coast of the Russian Caucasus is rich and well-studied (Table 5). None of the detected species is endemic to the Caucasus. All 62 species are widely distributed in Europe or even have transpalaearctic ranges.

Our study demonstrates that the biodiversity of a particular insect group in a particular region as well as its dynamics can be reconstructed using specimens deposited in museums and private collections like a picture from puzzle pieces.

### 4.2. History of Biodiversity

Thirty-four species were recorded before 1930, i.e., in the early period of the resort development. Twenty-nine of these species still occurred in the region in 1990–2020. Therefore, landscape transformation and intensive recreational use for several decades have not caused the large-scale extinction of native ladybird species, corresponding to the fact that many ladybird species successfully adapt to urbanization [46]. 

The general Coccinellidae biodiversity has not decreased. In contrast, several new ladybird species have appeared in each decade. Most of these new species became established and were collected in subsequent years. 

This detection of new species could not be explained by just the intensification of the study. For example, the average intensity of the ladybird collection in the region in the 1900s–1930s was 1.95 species per year, and the average intensity of collection was much lower in the 1940s–1970s—just 0.88 species per year. However, seven species were first recorded in the 1940s–1970s. If the new species had not appeared in the region, the number of newly-discovered species would have become lower in each subsequent period. In other words, if the species did not appear, it would become increasingly difficult to collect previously-undetected species in the same habitats using the same methods. However, in fact, the number of newly-detected species did not decrease with time, indicating that new species appeared in the region.

Of course, some species, especially rare ones, could have been overlooked by the collectors in the 1900s–1920s and were first detected later. However, the general tendency to increase Coccinellidae biodiversity cannot be explained by this situation since the main methods of collection were the same in all periods. This tendency indicates that new ladybird species appeared in the region. Most likely, the species could spread to the resort spontaneously or be unintentionally introduced with seedlings and other planting material or just a stowaway by transport. To reveal the reasons for the appearance or disappearance of the species in the region, the dynamics of the ranges should be studied in the future.

### 4.3. Populations of the Intentionally-Released Alien Coccinellidae Species

Our field studies have shown that the populations of *Rodolia cardinalis*, *Serangium montazerii, Cryptolaemus montrouzieri,* and *Lindorus lophanthae* still exist in the region. The first species is rare—only one restricted population in a tangerine grove was detected. The three other species occur both in this garden and on ornamental bushes in the streets of Sochi. None of these four alien species was detected outside the cities. Therefore, there are no reasons to suspect that they affect natural ecosystems. Nevertheless, the monitoring of the populations of these established alien ladybirds is necessary because the example of *Harmonia axyridis* has shown that the consequences of alien ladybird establishment could be unpredictable [12]. Special attention should be paid to the possibility of the spread of these alien species to Sochi National Park and other protected areas.

*Harmonia axyridis* has been released in the Caucasus many times since 1927. For example, in the 1980s, more than 107,000 specimens brought from the Far East were released in Georgia [36]. However, despite the massive releases, no established populations were found in the region before the 21st century. The first established population in the Caucasus was found in 2011 in Sochi [23]. Then, *H. axyridis* occupied the region quickly and became very abundant all over the Black Sea coast of the Russian Caucasus and in adjacent regions [9,20,47,48,49,50]. The dynamics of the range of *H. axyridis* in Europe clearly indicates that the species appeared in the Caucasus from the west as a result of the invasive range expansion [9,10,25]. However, since the release of specimens of *H. axyridis* from laboratory culture at the Black Sea coast of the Russian Caucasus continued at least until 2010 [51], it could not be excluded that the released specimens interbred with the specimens spontaneously invading the territory. Currently, *H. axyridis* has become the most common ladybird species in both cities and natural habitats, as has happened in many other regions of the world [12]. Monitoring the impact of this species on the Black Sea coast of the Russian Caucasus is advisable. It would also be interesting to observe which native species have become natural enemies of *H. axyridis* in the region and whether the parasites that appeared in the region as a result of coinvasion with *H. axyridis* [25] would infest native ladybird species.

### 4.4. The Impact of Establishment of H. axyridis

*Harmonia axyridis* has become very abundant in the region. However, the comparison of lists of ladybird species collected in different periods has shown that there is no evidence of the elimination of any species from the region after the establishment of *H. axyridis*.

The decline of native ladybirds caused by the establishment of *H. axyridis* was recorded in Belgium, Britain, Chile, and other countries [12,13,15]. The same tendency was recorded in the study of the ladybirds in the gardens in the center of Krasnodar Territory [52]. Our data do not contradict these results because we did not study the abundance of species but only the presence of the species in the particular year. In addition, the studies mentioned showed that the decline in ladybird biodiversity was detected in the selected plots (for example, in some alfalfa fields) [15]. Our results refer to regional biodiversity, and the available information for different habitats was used.

According to our observations, aphids are still very abundant in the streets of Sochi and other seaside towns of the region in spite of the large number of *H. axyridis*. It seems that the competition for food between aphidophagous species is not very strong, and the factor limiting the number of ladybirds is not starvation, but something else. Therefore, we suppose that it is unlikely that the establishment of *H. axyridis* at the Black Sea coast of the Caucasus would have a strong impact on competing local ladybird species by reducing their abundance as happened in some other regions. This conclusion is preliminary, since we have no quantitative data on aphid density, but only eye assessment.

On the other hand, not only competition, but indirect ecological interactions with *H. axyridis* could have a potential negative impact on other ladybird species connected with pathogens and parasites. For example, it is known that *Harmonia* can transmit parasitic microsporidia which are not harmful for it but could be lethal pathogens for other ladybirds [53]. The population of *H. axyridis* in the Caucasus is infected with the parasitic fungus *Hesperomyces virescens* (Ascomycota: Laboulbeniales) and the parasitic nematode *Parasitylenchus bifurcatus* (Nematoda: Tylenchida, Allantonematidae), which are probably alien to the Caucasus and entered the region as a result of co-invasion with their host [25]. Further observations are advisable to reveal if the native ladybirds would get infected with these parasites in the Caucasus.The population of *H. axyridis* in the region is young—approximately 10 years old. Further monitoring of Coccinellidae fauna of the region is advisable to determine the long-term consequences of *H. axyridis* invasion. The data integrated in our Appendix A could be used as a comparative basis for such monitoring in the future.

## 5. Conclusions

The ladybird fauna of the study region is rich and well-studied. 

Landscape transformation and intensive recreational use for almost 100 years have not caused large-scale extinction of native ladybird species.

Five alien species released in the region for pest control occur—*Rodolia cardinalis* was found in only one tangerine grove; *Cryptolaemus montrouzieri, Lindorus lophanthae,* and *Serangium montazerii* are more common and occur both in the gardens and in the streets of Sochi but were never found outside the city in natural habitats; *Harmonia axyridis* is usual both in anthropogenic and natural habitats.

Thirty-five ladybird species spontaneously appeared in the region between 1930 and 2020. 

There is no evidence of the elimination of any species from the region after the establishment of *H. axyridis*. 

Our study has shown that the biodiversity of a particular insect group in a particular territory can change rather quickly—in several decades. The traditional accumulative lists of regional insect fauna do not consider these changes. Therefore, the historical approach is necessary in studies of insect fauna. The general picture of the dynamics of fauna can be reconstructed using specimens deposited in museums and private collections like a picture from puzzle pieces.

## Figures and Tables

**Figure 1 insects-11-00824-f001:**
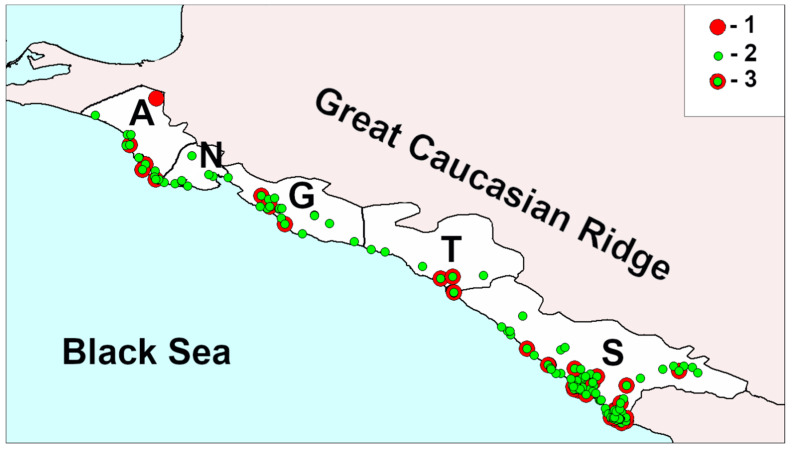
Region of study. A, Anapa; N, Novorossijsk; G, Gelendzhik; T, Tuapsinskiy district; S, Sochi; 1, localities of collection of *Harmonia axyridis;* 2, localities of collection of other ladybird species; 3, localities where both *H. axyridis* and other ladybird species were collected.

**Figure 2 insects-11-00824-f002:**
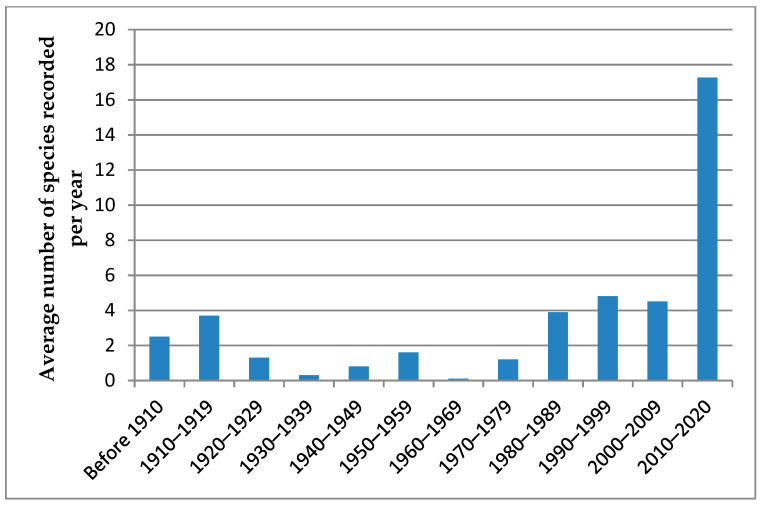
Intensity of ladybird collection in the region in different periods.

**Table 1 insects-11-00824-t001:** Species recorded before 1930. The species are arranged in the reverse chronological order of their first records. The decades when the species was detected are shaded with black. The species detected before 1930 but not detected later are shaded with pink.

Species	<1910	1910–1919	1920–1929	1930–1939	1940–1949	1950–1959	1960–1969	1970–1979	1980–1989	1990–1999	2000–2009	2010–2020
*Scymnus inderihensis* Mulsant, 1850												
*Exochomus quadripustulatus* (Linnaeus, 1758)												
*Ceratomegilla undecimnotata* (Schneider, 1792)												
*Harmonia quadripunctata* (Pontoppidan, 1763)												
*Coccinella septempunctata* Linnaeus, 1758												
*Coccinella hieroglyphica* Linnaeus, 1758												
*Adalia bipunctata* (Linnaeus, 1758)												
*Coccinella quinquepunctata* Linnaeus, 1758												
*Chilocorus renipustulatus* (Scriba, 1790)												
*Ceratomegilla notata* (Laicharting, 1781)												
*Coccinula sinuatomarginata* (Faldermann, 1837)												
*Bulaea lichatschovii* (Hummel, 1827)												
*Hippodamia tredecimpunctata* (Linnaeus, 1758)												
*Stethorus pusillus* (Herbst, 1797)												
*Hippodamia variegata* (Goeze, 1777)												
*Sospita vigintiguttata* (Linnaeus, 1758)												
*Coccinella magnifica* (Redtenbacher, 1843)												
*Calvia quatuordecimguttata* (Linnaeus, 1758)												
*Subcoccinella vigintiquatuorpunctata* (Linnaeus, 1758)												
*Psyllobora vigintiduopunctata* (Linnaeus, 1758)												
*Scymnus apetzi* Mulsant, 1846												
*Calvia quindecimguttata* (Fabricius, 1777)												
*Propylea quatuordecimpunctata* (Linnaeus, 1758)												
*Coccinula quatuordecimpustulata* (Linnaeus, 1758)												
*Hyperaspis campestris* (Herbst, 1783)												
*Scymnus auritus* Thunberg, 1795												
*Calvia decemguttata* (Linnaeus, 1767)												
*Platynaspis luteorubra* (Goeze, 1777)												
*Scymnus haemorrhoidalis* Herbst, 1797												
*Scymnus frontalis* (Fabricius, 1787)												
*Scymnus rubromaculatus* (Goeze, 1777)												
*Chilocorus bipustulatus* (Linnaeus, 1758)												
*Halyzia sedecimguttata* (Linnaeus, 1758)												
*Vibidia duodecimguttata* (Poda, 1761)												

**Table 2 insects-11-00824-t002:** The ladybird species appearing in the region during resort development. The decades when the species were detected are shaded with black. The species are arranged in the reverse chronological order of their first records.

Species	<1910	1910–1919	1920–1929	1930–1939	1940–1949	1950–1959	1960–1969	1970–1979	1980–1989	1990–1999	2000–2009	2010–2020
*Scymnus ater* Kugelann, 1794												
*Clitostethus arcuatus* (Rossi, 1794)												
*Scymnus interruptus* (Goeze, 1777)												
*Coccinella undecimpunctata* Linnaeus, 1758												
*Myrrha octodecimguttata* (Linnaeus, 1758)												
*Anisosticta novemdecimpunctata* (Linnaeus, 1758)												
*Nephus bipunctatus* (Kugelann, 1794)												
*Hyperaspis concolor* Suffrian, 1843												
*Tytthaspis sedecimpunctata* (Linnaeus, 1761)												
*Scymnus suturalis* Thunberg, 1795												
*Parexochomus melanocephalus* (Zoubkoff, 1833)												
*Oenopia conglobata* (Linnaeus, 1758)												
*Nephus quadrimaculatus* (Herbst, 1783)												
*Coccidula scutellata* (Herbst, 1783)												
*Myzia oblongoguttata* (Linnaeus, 1758)												
*Aphidecta obliterata* (Linnaeus, 1758)												
*Nephus redtenbacheri* (Mulsant, 1846)												
*Parexochomus nigromaculatus* (Goeze, 1777)												
*Oenopia impustulata* (Linnaeus, 1767)												
*Adalia decempunctata* (Linnaeus, 1758)												
*Scymnus subvillosus* (Goeze, 1777)												
*Hyperaspis reppensis* (Herbst, 1783)												
*Hyperaspis femorata* Motschulsky, 1837												

**Table 3 insects-11-00824-t003:** The ladybird species intentionally released for pest control and established in the region. The decades when the releases of the species began are shaded with black and indicated with “i”. The decades when the species were detected by us are shaded with black.

Species	<1910	1910–1919	1920–1929	1930–1939	1940–1949	1950–1959	1960–1969	1970–1979	1980–1989	1990–1999	2000–2009	2010–2020
*Harmonia axyridis* (Pallas, 1773)			i									
*Rodolia cardinalis* (Mulsant, 1850)				i								
*Cryptolaemus montrouzieri* (Mulsant, 1853) 1853)				i								
*Lindorus lophanthae* (Blaisdell, 1892)					i							
*Serangium montazerii* (Fürsch, 1995)								i				

**Table 4 insects-11-00824-t004:** Ladybird species collected in 1985–2004 and in 2016–2020. Aphidophagous species are marked with an asterisk (*).

In 1985–2004	In Both Periods	In 2016–2020
*Bulaea lichatschovii* *Calvia quindecimguttata ** *Ceratomegilla undecimnotata ** *Coccidula scutellata ** *Hippodamia tredecimpunctata ** *Hyperaspis concolor* *Myzia oblongoguttata ** *Oenopia impustulata ** *Parexochomus melanocephalus **	*Adalia bipunctata ** *Adalia decempunctata ** *Calvia decemguttata ** *Calvia quatuordecimguttata ** *Chilocorus bipustulatus ** *Chilocorus renipustulatus ** *Coccinella septempunctata ** *Coccinula quatuordecimpustulata ** *Halyzia sedecimguttata ** *Harmonia quadripunctata ** *Hippodamia variegate ** *Hyperaspis femorata* *Nephus bipunctatus* *Nephus quadrimaculatus* *Nephus redtenbacheri ** *Oenopia conglobate ** *Parexochomus nigromaculatus ** *Platynaspis luteorubra ** *Propylea quatuordecimpunctata ** *Psyllobora vigintiduopunctata* *Scymnus apetzi ** *Scymnus frontalis ** *Scymnus haemorrhoidalis ** *Scymnus rubromaculatus ** *Scymnus subvillosus ** *Scymnus suturalis ** *Stethorus pusillus* *Subcoccinella vigintiquatuorpunctata* *Tytthaspis sedecimpunctata ** *Vibidia duodecimguttata **	*Clitostethus arcuatus ** *Coccinella magnifica ** *Coccinella quinquepunctata ** *Exochomus quadripustulatus ** *Hyperaspis reppensis ** *Myrrha octodecimguttata ** *Scymnus ater ** *Scymnus auritus ** *Scymnus interruptus **

**Table 5 insects-11-00824-t005:** Biodiversity of Coccinellidae in some regions of the Caucasus.

Region	Area (km^2^)	Number of Recorded Coccinellidae Species	Sources of Information
Black Sea coast of Russia	8913	62	Current study
Adygea	7800	30	[43]
Dagestan	50,300	27	[44]
Georgia	69,700	85	[45]

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
