# Peer review of "History of the Biodiversity of Ladybirds (Coccinellidae) at the Black Sea Coast of the Russian Caucasus in the Last 120 Years—Does the Landscape Transformation and Establishment of Harmonia axyridis Have an Impact?"

_insects, 2020, doi:10.3390/insects11110824_

Round 1

Reviewer 1 Report

  1. The references can be improved through publications: 1) about the Harmonia - Ruchin A.B., Egorov L.V., Lobachev E.A., Lukiyanov S.V., Sazhnev A.S., Semishin G.B. 2020. Expansion of Harmonia axyridis (Pallas, 1773) (Coleoptera: Coccinellidae) to European part of Russia in 2018–2020. Baltic J. Coleopterol., 20 (1): 51 – 60. 2) about changes in the nature of the Caucasus Tuniyev B.S., Timukhin I.N. 2017. Species composition and comparative-historical aspects of expansion of alien species of vascular plants on the Sochi Black Sea Coast (Russia). Nature Conservation Research 2(4): 2–25. http://dx.doi.org/10.24189/ncr.2017.046 and Bitukov N.A., Shagarov L.M. 2017. Degradation of water protection function of the Western Caucasus mountain oakeries as a result of fellings. Nature Conservation Research 2(3): 40–47. http://dx.doi.org/10.24189/ncr.2017.006
  2. In the citations of publications there are many links to Internet resources. It seems to me that it is advisable to reduce them.
  3. Materials and Methods. One paragraph (lines 132-139) is superfluous. It should be moved to the Discussion.

Author Response

Thank you very much for the quick review and valuable comments.

  1. The references can be improved through publications: 1) about the Harmonia - Ruchin A.B., Egorov L.V., Lobachev E.A., Lukiyanov S.V., Sazhnev A.S., Semishin G.B. 2020. Expansion of Harmonia axyridis (Pallas, 1773) (Coleoptera: Coccinellidae) to European part of Russia in 2018–2020. Baltic J. Coleopterol., 20 (1): 51 – 60. 2) about changes in the nature of the Caucasus Tuniyev B.S., Timukhin I.N. 2017. Species composition and comparative-historical aspects of expansion of alien species of vascular plants on the Sochi Black Sea Coast (Russia). Nature Conservation Research 2(4): 2–25. http://dx.doi.org/10.24189/ncr.2017.046 and Bitukov N.A., Shagarov L.M. 2017. Degradation of water protection function of the Western Caucasus mountain oakeries as a result of fellings. Nature Conservation Research2(3): 40–47. http://dx.doi.org/10.24189/ncr.2017.006

  • Thank you for this comment. The first and the second references have been added. The third one seems not to be relevant to our article.

  1. In the citations of publications there are many links to Internet resources. It seems to me that it is advisable to reduce them.

  • The link to the Internet resource on the history of Sochi is changed to the reference to the book. We have kept other links since we believe them to be necessary. For example, the first record of the established population of H. axyridis in the Caucasus was published by T. Mogilevich on-line.

  1. Materials and Methods. One paragraph (lines 132-139) is superfluous. It should be moved to the Discussion.
  • It is moved to the Discussion. Thank you very much for this good advice.

Reviewer 2 Report

The manuscript presents an interesting analysis of the changes in the ladybird beetle species richness at the Black Sea coast of Russia over some 120 years. The authors relate these changes to the landscape transformation and the appearance of alien ladybird species, especially the harlequin ladybird. The main conclusions from the study are that (1) the landscape transformation (anthropogenization) has led to an increase of the species richness rather than to its decrease and (2) there is no evidence of the effect of alien species on the extinction of native ladybirds.

Some minor remarks and suggestions are listed below:

line 14: ‘it is necessary to monitor ...’ – I would express this less categorically, e.g. ‘it is advisable to monitor ...’.

line 15: I would change ‘We analyzed the dynamics of fauna at the main sea resort of Russia for 120 years’ to ‘We analyzed the dynamics of the fauna at the main sea resort of Russia over a period of 120 years’.

line 27: change ‘for 120 years’ to ‘over a period of 120 years’.

line 32: change ‘from 2011-2020’ to ‘in recent years (2011-2020)’.

line 35: change ‘from 1930-2020’ to ‘between 1930 and 2020’.

line 50: ‘Intentional monitoring’ – I wonder whether monitoring can be unintentional.

line 170: I would change ‘from 1990-2020’ to either ‘in recent times (1990-2020)’ or ‘between 1990 and 2020’ or ‘from 1990 to 2020’.

line 177: ‘grain fields of Anapa’ - Unclear why only the grain fields of Anapa are mentioned. Are the grain fields absent from other parts of the study area? Above (line 174), three district are mentioned.

lines 181-182: I would change the sentence ’Only one record of each species is known, indicating that these species were probably rare in the region’ to ‘Only one record of each of these species indicates that they were probably rare in the region’.

line 193: I would not start a new paragraph here but continue the previous one.

line 202: The sentence ‘The average number of first records is three new species per decade’ needs to be reworded. Maybe ‘In the study area, the average rate of recording new ladybird species is three per decade’ ?

line 208: I would change ’the species were then repeatedly collected’ to ’ many of the species in this group were then repeatedly collected’.

lines 209-210: I would change ’ collected for 11 years from 1957-2020’ to ‘collected in 11 years during the period 1957-2020’.

line 216: change ’from 1970-2020’ to ’between 1970 and 2020’.

line 227: Chilocorus kuwanae should not be listed here or at least shouldn’t be listed under this name. It was synonymized with C. renipustulatus by the first author of this paper [34], and Table 1 indicates that C. renipustulatus is a constant, native element of the ladybird fauna at the Black Sea coast of Russia.

Table 4: Hyperaspis concolor is listed twice, in the first and second column of the table.

lines 292-294: change the font in the ladybird names to italics.

lines 295-296: ’ they could be absent only occasionally in the collections in 2016-2020’ – this is totally unclear to me.

lines 314-315: ‘All 62 ladybird species were recorded, ...’ – I suppose that the authors intended to write ‘Altogether’ or ‘In total’ instead of ‘All’.

line 356: change ‘interbred with the species ...’ to ‘interbred with the specimens ...’

line 379: change ‘are rich’ to ‘is rich’.

line 389: change ‘from 1930-2020’ to ‘between 1930 and 2020’.

In my opinion, the last section of the MS (Conclusions) is superfluous. With the exception of the last paragraph of this section, it repeats the main results rather than presents the conclusions drawn from the study. The statements included in the last paragraph (lines 395-398) may be transferred to the Discussion.

Author Response

 Thank you very much for the quick review, the attentive reading of our article and very good advises.

line 14: ‘it is necessary to monitor ...’ – I would express this less categorically, e.g. ‘it is advisable to monitor ...’.

  • “necessary” is replaced by “advisable”

line 15: I would change ‘We analyzed the dynamics of fauna at the main sea resort of Russia for 120 years’ to ‘We analyzed the dynamics of the fauna at the main sea resort of Russia over a period of 120 years’.

  • “We analyzed the dynamics of fauna at the main sea resort of Russia for 120 years” is changed to “We analyzed the dynamics of thefauna at the main sea resort of Russia over a period of 120 years”

line 27: change ‘for 120 years’ to ‘over a period of 120 years’.

  • “for 120 years” is changed to “over a period of 120 years”

line 32: change ‘from 2011-2020’ to ‘in recent years (2011-2020)’.

  • “from 2011-2020” is changed to “in recent years (2011-2020)”

line 35: change ‘from 1930-2020’ to ‘between 1930 and 2020’.

  • “from 1930-2020” ” is changed to “between 1930 and 2020”

line 50: ‘Intentional monitoring’ – I wonder whether monitoring can be unintentional.

  • “Unintentional monitoring” would be funny. “intentional” is deleted.

line 170: I would change ‘from 1990-2020’ to either ‘in recent times (1990-2020)’ or ‘between 1990 and 2020’ or ‘from 1990 to 2020’.

  • We changed “from 1990-2020” to “between 1990 and 2020”

line 177: ‘grain fields of Anapa’ - Unclear why only the grain fields of Anapa are mentioned. Are the grain fields absent from other parts of the study area? Above (line 174), three district are mentioned.

  • We deleted the hypothesis about grain fields. Coccinula sinuatomarginata is just a rare species.

lines 181-182: I would change the sentence ’Only one record of each species is known, indicating that these species were probably rare in the region’ to ‘Only one record of each of these species indicates that they were probably rare in the region’.

  • “Only one record of each species is known, indicating that these species were probably rare in the region” is changed to Only one record of each of these species indicates that they were probably rare in the region

line 193: I would not start a new paragraph here but continue the previous one.

  • The paragraphs are merged.

line 202: The sentence ‘The average number of first records is three new species per decade’ needs to be reworded. Maybe ‘In the study area, the average rate of recording new ladybird species is three per decade’ ?

  • “The average number of first records is three new species per decade” is changed to “In the study area, the average rate of recording new ladybird species is three per decade”

line 208: I would change ’the species were then repeatedly collected’ to ’ many of the species in this group were then repeatedly collected’.

  • Thank you. “the species were then repeatedly collected” is replaced by “many of the species in this group were then repeatedly collected”

lines 209-210: I would change ’ collected for 11 years from 1957-2020’ to ‘collected in 11 years during the period 1957-2020’.

  • “collected for 11 years from 1957-2020” is replaced by “collected in 11 years during the period 1957-2020”

line 216: change ’from 1970-2020’ to ’between 1970 and 2020’.

  • “from 1970-2020” is replaced by “between 1970 and 2020”.

line 227: Chilocorus kuwanae should not be listed here or at least shouldn’t be listed under this name. It was synonymized with C. renipustulatus by the first author of this paper [34], and Table 1 indicates that C. renipustulatus is a constant, native element of the ladybird fauna at the Black Sea coast of Russia.

  • Yes! Thank you so much! We deleted Chilocorus kuwanaefrom the list and added the explanation: “Besides that, specimens of “ kuwanae Silvestri, 1909” introduced from Japan were released [5], but we have revealed that C. kuwanae Silvestri, 1909 is the junior synonym of C. renipustulatus (Scriba, 1790), which is the native element of the ladybird fauna at the Black Sea coast of Russia [40].”

Table 4: Hyperaspis concolor is listed twice, in the first and second column of the table.

  • Many thanks for your attention! Hyperaspis concolor is deleted in the second column.

lines 292-294: change the font in the ladybird names to italics.

  • The font is changed.

lines 295-296: ’ they could be absent only occasionally in the collections in 2016-2020’ – this is totally unclear to me.

  • We deleted it. We just mean that these species are rare in the region.

lines 314-315: ‘All 62 ladybird species were recorded, ...’ – I suppose that the authors intended to write ‘Altogether’ or ‘In total’ instead of ‘All’.

  • We changed “all” to “In total”.

line 356: change ‘interbred with the species ...’ to ‘interbred with the specimens ...’

  • “species” is changed to “specimens”

line 379: change ‘are rich’ to ‘is rich’.

  • “are rich” is changed to “is rich”

line 389: change ‘from 1930-2020’ to ‘between 1930 and 2020’.

  • It is changed.

In my opinion, the last section of the MS (Conclusions) is superfluous. With the exception of the last paragraph of this section, it repeats the main results rather than presents the conclusions drawn from the study. The statements included in the last paragraph (lines 395-398) may be transferred to the Discussion.

  • The section “Conclusions” is obligatory according to the Authors Guidelines. We made it more concise to avoid repetitions with the “Discussion” section.

Thank you so much for the very attentive reading of our manuscript!

Your advises were really helpful!

Reviewer 3 Report

This interesting study can be published after appropriate revision.

The first problem is that the title should be changed because it does not correspond to the content of the study. "History of biodiversity of the ladybirds" assumes a reconstruction of Holocene events, not 120 years. I suggest a more correct name "Change fauna of the ladybirds (Coccinellidae) at the Black Sea coast of Russia for 120 years: do the landscape transformation and establishment of Harmonia axyridis have an impact?".

The second problem is the lack of collection according to a specific method for 120 years. This makes it impossible to adequately assess changes in fauna. Unfortunately I don't know how to solve this problem, I would not write such article because of the high degree of assumptions. 

Third problem. I do not understand the reason for the extinction of some species. Authors should discuss this in detail.

Fourth problem. Tables 1-3 are completely unacceptable to the reader. They need to be redone.

Author Response

Thank you very much for the quick review and valuable conceptual questions! They helped us to explain our approach better.

The title should be changed because it does not correspond to the content of the study. "History of biodiversity of the ladybirds" assumes a reconstruction of Holocene events, not 120 years. I suggest a more correct name "Change fauna of the ladybirds (Coccinellidae) at the Black Sea coast of Russia for 120 years: do the landscape transformation and establishment of Harmonia axyridis have an impact?".

  • We have changed the title and re-written the beginning of the introduction to make our concept clearer. The historical approach, i.e., the study of current dynamics of floras and faunas is now becoming more and more popular in zoology and botany. The accumulated information on localities of species detection (in particular, in Global Biodiversity Information Facility) clearly indicates that the ranges of plant and animal species are changing quickly: it is proved that significant changes often occur in just several decades. Since the species live in the territories which are intensively used and changed by humans for many centuries, we believe that the current regional floras and faunas are the product of the long-term interactions between nature and human society. Therefore, the history of biodiversity should be analyzed in the context of human history.

The second problem is the lack of collection according to a specific method for 120 years. This makes it impossible to adequately assess changes in fauna. Unfortunately I don't know how to solve this problem, I would not write such article because of the high degree of assumptions. 

  • Fortunately, the methods of ladybird collection were the same in the 19th, 20th and 21st The old museums have accumulated enough specimens collected in all periods. So the data obtained in different periods are comparable. We do not agree that there is no collection according a specific method for 120 years. Coccinellidae is a very popular group and the Black Sea Coast of the Caucasus is the main sea resort of Russia, so it is not surprising, that many entomologists visited this region and collected ladybirds there. In particular, we found approximately two hundred specimens collected in the late 19th and early 20th centuries in the collections of the Zoological Institute of Russian Academy of Sciences (St.-Petersburg) and Zoological Museum of Moscow State University (Moscow). Thousands of specimens were collected in the subsequent periods and deposited into different museums and private collections. Therefore, we have enough material to reconstruct the changes of the ladybird fauna of the region during the last century. The degree of assumptions in studies on the changes of the regional faunas during 100 years is definitely much less than the degree of assumption in studies on a reconstruction of Holocene events.

Third problem. I do not understand the reason for the extinction of some species. Authors should discuss this in detail.

  • When we began our work, we supposed that some species which occurred in the region before the development of the resort (before 1930) would become extinct later because of the destruction of their habitats. But surprisingly this hypothesis was not confirmed. Twenty-nine of 34 species recorded before 1930 were also recorded in the region between 1990 and 2020. Only five species recorded before 1930 were not recorded recently, but each of these species was found only once in the history. So it is unknown if they are “extinct” or just rare. They could be found in the region once again in future, as it happened with Coccinella magnifica: it was found in 1907 and in 2016.

Fourth problem. Tables 1-3 are completely unacceptable to the reader. They need to be redone.

  • The specific problem with the tables is not indicated, and other three reviewers like our tables. So we decided not to change the tables.

Reviewer 4 Report

Dear Andrzej & Marina,

your manuscript is, in my opinion, an excellent study of high importance, it should be published in the journal you chose. It provides useful information for nature conservation decision makers as well as interesting methodological aspects. Label data from collection specimens are a grossly underrated possible source of faunistic and ecological information.

I have only few suggestions to make:

(1) You should give the years in which the additionally mentioned species have been intentionally released.

(2) What is the source for the statement that Harmonia axyridis is used since about 100 years for pest control?

(3) In the Results section, you mix presentation and discussion of results. I recommend to shift the discussions to the Discussion section.

(4) You correctly state that there are no data available on abundancies. Nevertheless, it would be worth discussing or at least mentioning in some detail the possibility that H.axyridis might have a strong ecological impact on competing local species by reducing their abundance or by urging them to shift to different strata.

(5) It would be a service to the reader and would facilitate a more informed judgement on the relevance of your study if you would mark or highlight which species are possible competitors of H.axyridis and which not. The probability that non-aphidophagous species are affected by the spread of H.axyridis is nearly zero.

I added some minor comments on the attached PDF.

I do not insist in anonymity. You are welcome to contact me if you wish.

Michael Schmitt ([email protected])

Author Response

Dear Michael,

Thank you very much for the quick review, for your kind words about our work and for your valuable comments.

(1) You should give the years in which the additionally mentioned species have been intentionally released.

  • The years are added.

(2) What is the source for the statement that Harmonia axyridis is used since about 100 years for pest control?

  • We added the reference to the review by Roy et al., 2016. The intentional releases of H. axyridis began in California in 1916.

(3) In the Results section, you mix presentation and discussion of results. I recommend to shift the discussions to the Discussion section.

  • You are right. We deleted some discussion paragraphs from the results, since the same information is already present in Discussion section and transferred some paragraphs from “Results” to “Discussion”.

(4) You correctly state that there are no data available on abundancies. Nevertheless, it would be worth discussing or at least mentioning in some detail the possibility that H.axyridis might have a strong ecological impact on competing local species by reducing their abundance or by urging them to shift to different strata.

  • The situation with the possible impact of H. axyridis on other ladybird species is more complex. Aphids are very abundant in the streets of Sochi resort. The plantings of this city are entirely artificial and consist of plants introduced from over the world. It is the real heaven for aphids. We don’t feel that the aphidophagous ladybirds can starve there. It seems that there are enough (more than enough) aphids to eat in spite of large number of H. axyridis. And the factor limiting the number of ladybirds is not the starvation, but something else.

 (5) It would be a service to the reader and would facilitate a more informed judgement on the relevance of your study if you would mark or highlight which species are possible competitors of H.axyridis and which not. The probability that non-aphidophagous species are affected by the spread of H.axyridis is nearly zero.

- We added the information about what species are aphidophagous to the Table 4. But Harmonia axyridis can affect both aphidophagous  and non-aphidophagous by indirect impact connected with pathogens and parasites. At least two alien parasitic species entered the Caucasus together with their host H. axyridis: the nematode living in the body cavity and the ectoparasitic fingus [https://journals.plos.org/plosone/article?id=10.1371/journal.pone.0202841]. In September 2020 we found this fungus to infect the mycetophagous ladybird Psyllobora vigintiduopunctata in Sochi. This finding is not included to the manuscript, since we are preparing a separate communication on this finding with another author. But we added some discussion about these parasites to the manuscript.

(6) how do you define "top predator"? The harlequin ladybird is most probably not a "predator at the top of a food chain, without natural predators" (Wikipedia)

- “Top predator” is deleted. Other minor corrections proposed by you are done.

Best wishes,

Marina and Andrzej

Round 2

Reviewer 3 Report

I verified that the authors corrected and improved the text.

Author Response

  • I verified that the authors corrected and improved the text.

Thank you very much.